# Men’s Psychotherapy Use, Male Role Norms, and Male-Typical Depression Symptoms: Examining 716 Men and Women Experiencing Psychological Distress

**DOI:** 10.3390/bs11060083

**Published:** 2021-06-02

**Authors:** Lukas Eggenberger, Callia Fordschmid, Claudio Ludwig, Seraina Weber, Jessica Grub, Nikola Komlenac, Andreas Walther

**Affiliations:** 1Department of Clinical Psychology and Psychotherapy, Psychological Institute, University of Zurich, CH-8050 Zurich, Switzerland; lukas.eggenberger@uzh.ch (L.E.); callia.fordschmid@uzh.ch (C.F.); claudio.ludwig@uzh.ch (C.L.); seraina.weber@intergga.ch (S.W.); j.grub@psychologie.uzh.ch (J.G.); 2Gender Medicine Unit, Medical University of Innsbruck, A-6020 Innsbruck, Austria; nikola.komlenac@i-med.ac.at

**Keywords:** depression, male role norms, masculinity, help-seeking, psychotherapy, gender medicine, MDRS-22, PHQ-9

## Abstract

Men as compared to women are half as often affected by depressive and anxiety disorders and seek significantly less help for mental health issues than women. Adherence to traditional male role norms (AtTMRN) may hinder men from describing prototypical depression symptoms and from seeking psychotherapy. The current study compared whether AtTMRN, gender role identity, or the experience of prototypical or male-typical externalizing mental health symptoms were associated with psychotherapy use in men and women. In an anonymous online survey, 716 participants (37% men) reporting to currently experience psychological distress were examined. Information was obtained on psychotherapy use, depression and anxiety symptoms, gender role identity, and traditional male role norms. Although experiencing similar levels of depression, men compared to women showed a reduction in psychotherapy use by 29%. Masculine role identity was directly associated with reduced psychotherapy use in men (β = −0.41, *p* = 0.029), whereas AtTMRN was not (men: β = −0.04, *p* = 0.818; women: β = −0.25, *p* = 0.064). Higher externalizing depression symptomatology (β = −0.68, *p* = 0.005), but not prototypical depression symptomatology (β = −0.02, *p* = 0.499), was associated with reduced psychotherapy use in men but not women (*p* > 0.05). Interactions revealed that men, but not women, with high AtTMRN use psychotherapy only when exhibiting elevated symptom levels. The results corroborate previous reports showing reduced psychotherapy use in men as compared to women and identify elevated masculine role identity and male-typical externalizing depression symptomatology as direct factors associated with reduced psychotherapy use in psychologically distressed men. AtTMRN interacts with mental health symptoms to predict psychotherapy use, indicating that men with high AtTMRN only use psychotherapy when exhibiting high symptomatology.

## 1. Introduction

Men as compared to women have approximately a 30% lower utilization of psychiatric and psychotherapeutic services [1,2]. A nationally representative sample from Germany comprised of 24,016 adults revealed that 8.1% men but 11.2% women used any form of outpatient psychiatric or psychotherapeutic service in the past year [3]. This gender difference seems to be even more pronounced in patients with depression, where men are only half as likely to start psychotherapy compared to women [4,5]. Whereas, in Australia encouraging trends have been reported with increased use of specialized mental health services by 92.7% in men between the years 2007 and 2012 [6], no such amelioration of the general psychiatric or psychotherapeutic outpatient service use among men can be observed in European countries. By contrast, in a large sample from Germany examining help-seeking for mental disorders over a 12-year period, a significant reduction in women who do not seek help was identified, but no difference in help-seeking behavior in men was observable [7]. This indicates a further aggravation of the gender gap in help-seeking for mental disorders in German-speaking Europe.

However, since psychiatric or psychotherapeutic service use is entirely dependent on experiencing mental health problems, one could speculate that men exhibit less mental health problems and therefore need less psychotherapy than women. Indeed, depressive and anxiety disorders are the most prevalent mental health disorders worldwide and men seem to suffer from depression only half as often as women [8,9]. Similar differences in prevalence rates are observed for anxiety disorders [10]. Different biologically based perspectives exist to explain gender differences in prevalence rates of depressive and anxiety disorders [11,12,13,14,15]. However, these approaches can explain gender differences only to a certain extent. Research increasingly supports the notion that the presentation of male-typical externalizing depression and anxiety phenotypes significantly contributes to gender differences in depression and anxiety prevalence rates as well as help-seeking for mental health problems.

In line with this, an emerging body of research suggests that gender differences in prevalence rates of depression vanish when taking prototypical symptoms like anhedonia, depressive mood, fatigue, and male-typical externalizing symptoms such as anger and aggression, risky behavior, or substance abuse into account [16,17,18]. Early on, researchers hypothesized that there might be a percentage of men suffering from depressive or anxiety disorders but exhibiting male-typical phenotypes, which are not captured by commonly used diagnostic criteria and screening tools [19]. Consequently, because of their aberrant symptomatology, these men often do not engage psychiatric or psychotherapeutic services. With the development of male-specific depression screenings, researchers started identifying and better describing this portion of men suffering from psychological distress and exhibiting externalizing symptoms [1,18,20,21].

Male gender socialization and adherence to traditional male role norms (AtTMRN) are suggested to underlie this male typical expression of psychological distress via externalizing symptoms [22,23,24]. Men experiencing psychological distress seem to perceive it as a threat for their masculinity self-concept to present prototypical (masculinity self-concept incompatible) depression symptoms or to engage in psychotherapy. By being depressed or engaging in psychotherapy, they assume to be regarded as weak, feminine, vulnerable, or inferior [25]. This threat perception leads men to present masculinity self-concept compatible externalizing symptoms, reinforcing reluctance towards psychotherapy and rendering it more difficult for clinicians to correctly assess a man’s mental condition.

Thus, reported gender differences in prevalence rates for depression and anxiety disorders as well as psychotherapy use may be founded on masculine role identity and male role norms [1,26,27]. Gender role identity refers to a person’s belief of whether they possess certain personality characteristics traditionally viewed as either masculine or feminine [28]. According to the gender role identity paradigm, personality characteristics that are typically ascribed to be masculine are strength, rationality, ambitiousness, independence, and leadership. On the other hand, personality characteristics that are typically ascribed to be feminine are compassion, emotionality, sensitivity, and dependence [29]. Masculine gender role identity, as measured with the Bem Sex Role Inventory (BSRI), has been associated with higher self-esteem, self-confidence, and reduced depression and anxiety symptoms [30,31,32]. Yet, it has been reported that observed help-seeking was not associated with perceived masculinity [33].

In contrast to the gender role identity paradigm, the gender-role strain paradigm stresses the fact that no person can embody all personality characteristics that are traditionally ascribed to women or men. The idealized male or female personality characteristics are based on socially defined gender role norms of how members of a particular gender should be and behave [34,35]. Traditional male role norms depict men as independent, dominant, stoic, strong, self-reliant, and not emotional [35]. The gender-role strain paradigm further highlights that AtTMRN across all situations can lead to stress and therefore to negative psychological consequences [28,36]. Accordingly, a meta-analytic examination showed that conformity to traditional male role norms is associated with worse mental health outcomes as well as with worse attitudes toward help-seeking in men [37]. Yousaf and colleagues [38] further showed a negative correlation indicating that higher AtTMRN is associated with worse attitudes toward seeking psychological help in men but not women. Further research based on male samples showed associations between AtTMRN and psychological help-seeking intentions [39], attitudes towards seeking psychological help [40], a willingness to seek help for mental health issues [41], and a reduced frequency of mental health service visits [42]. 

This line of research supports the perspective that the internalization of male role norms (e.g., the need to be self-reliant) presents one major reason hindering men to engage in psychotherapy. However, a limitation of previous studies is that only the attitude or intention to start psychotherapy was measured but not the actual psychotherapy use [40,41,42]. Additionally, most of the studies investigated college student samples and did not sufficiently include individuals with psychological distress or control for it, suggesting the limited generalizability of previous studies to the population of psychologically distressed individuals. Furthermore, most studies did not exclusively examine psychotherapy use, but mental health service use in general, which introduces imprecision to the question whether men with a high AtTMRN take on psychotherapeutic services less than men with a low AtTMRN.

Psychotherapy use may further be predicted by the interaction between AtTMRN and mental health symptoms. Notably, previous research showed that men and women with higher AtTMRN score higher on the Male Depression Risk Scale (MDRS-22) and a widely used depression measure assessing prototypical depression symptoms (PHQ-9) [43]. Men with a high AtTMRN exhibited particularly elevated MDRS-22 scores, suggesting a potential interaction of those constructs with regard to psychotherapy use [43]. Thus, men with a high AtTMRN suffering from depression or anxiety might present predominantly masculinity-compatible mental health symptoms and concomitantly experience a conflict with regard to psychotherapy use, where they would need to admit to needing help, thereby revealing emotional vulnerability. However, in order to comply with internalized role norms (e.g., self-reliance, toughness, anti-femininity), men with a high degree of AtTMRN who exhibit high levels of externalizing mental health symptoms may refuse to seek psychotherapy for a relatively long time in the course of their mental illness and may only start psychotherapy later during the disease course when the situation becomes unbearable with more severe symptomatology [44]. A consequence of not seeking help or seeking help at a later point during the disease course may be an increased risk for suicidal behavior in men [45].

Taken together, men use psychiatric and psychotherapeutic services considerably less than women, which might be related to the presentation of male-typical mental health symptoms and the underlying AtTMRN and their interplay. Although there is an extensive body of research examining AtTMRN and attitudes toward help-seeking [37,38,39,40,41], studies focusing on actual psychotherapy use in psychologically distressed men are lacking. The present study aimed to contribute to this important, yet still insufficiently explored research area by concomitantly analyzing prototypical and externalizing depression and anxiety symptomatology, gender role identity, AtTMRN, and their interactions as predictors of actual use of psychotherapy in a sample of men and women self-reporting to suffer from psychological distress.

## 2. Materials and Methods

### 2.1. Procedure and Sample 

This anonymous, cross-sectional study was approved by the ethical review board of the Faculty of Arts and Social Sciences of the University of Zurich (Nr. 20.2.10). For the purpose of participant recruitment, the study was distributed through advertisements on social media platforms such as Facebook across German-speaking countries in Europe (Switzerland, Germany, Austria, Liechtenstein, and Luxembourg) and the study’s webpage. The aim was to recruit a diverse sample of men and women with respect to sociodemographic and psychological variables such as age, self-reported psychological distress, and psychotherapy use to increase the external validity of the findings. Participants had to be older than 18 years old and have good knowledge of the German language. A total of 2150 people expressed interest in this study during the recruitment period from March until May 2020; 1303 of the participants were not included in the final analyses for one of the following reasons: missing consent (women *N* = 3, missing gender *N* = 268), self-reported insufficient German knowledge to complete questionnaires (women *N* = 3, men *N* = 2, missing gender *N* = 114), not experiencing psychological distress at the time of survey completion (women *N* = 348, men *N* = 326), age below 18 (men *N* = 3), missing or unrealistic age (women *N* = 1, men *N* = 4), and/or incomplete data in any of the relevant questionnaires (women *N* = 192, men *N* = 126) (Figure 1). This led to a total of 716 participants. 

### 2.2. Instruments

#### 2.2.1. Sociodemographics, Self-Reported Psychological Distress and Psychotherapy Use 

Participants received, at the beginning of the survey, different sociodemographic questions such as age, binary gender (“male,” “female”), dimensional gender (positioning between the two poles 1 = “masculine,” 10 = “feminine”), education (“none completed,” “secondary education,” “tertiary education,” “other”) relationship status (“single,” “in a relationship,” “married,” “in a registered partnership,” “divorced,” “widowed”), and sexuality (“heterosexual,” “homosexual,” “bisexual,” “asexual,” “other”). Further, participants were asked to indicate by self-report whether they were currently experiencing psychological distress (“yes,” “no”) and whether they were currently undergoing psychotherapeutic treatment (“yes,” “no”). 

#### 2.2.2. Patient Health Questionnaire-9 

The Patient Health Questionnaire-9 (PHQ-9) [46] assesses nine major depressive symptoms specified by the Diagnostic and Statistical Manual of Mental Disorders (DSM-5) [47] within the preceding two-week period in self-reporting a total of nine items with a four-point Likert scale ranging from 0 (“not at all”) to 3 (“almost every day”). Well validated (Cronbach’s α = 0.86–0.89) [46], the PHQ-9 is commonly used for assessing depression severity and case finding of major depressive disorder with a cut-off ≥ 10 within research and clinical practice [48]. A German version of the PHQ-9, which has been previously validated on a representative German-speaking sample, was used in the present study (Cronbach’s α = 0.89) [49]. Cronbach’s α in the current study was at 0.87 for men and 0.85 for women.

#### 2.2.3. Male Depression Risk Scale-22 

The Male Depression Risk Scale-22 (MDRS-22) [43] assesses externalizing depressive symptoms within the preceding month. It consists of 22 items with an eight-point Likert scale ranging from 0 (“not at all”) to 7 (“almost always”). The validated German version of the MDRS-22 (Guttman’s λ_2_ = 0.62–0.91) [18] was used in this present study. Cronbach’s α in the current study was at 0.85 for men and 0.84 for women.

#### 2.2.4. Generalized Anxiety Disorder-7 

The Generalized Anxiety Disorder-7 (GAD-7) [50] includes seven self-reported items assessing the presence and severity of generalized anxiety disorder with a four-point Likert scale ranging from 0 (“not at all”) to 3 (“nearly every day”). The German version [51] showed good validity and reliability (Cronbach’s α = 0.85) in a representative German population sample [52]. Cronbach’s α in the current study was at 0.88 for men and 0.85 for women.

#### 2.2.5. Bem Sex-Role Inventory 

The Bem Sex-Role Inventory (BSRI) [53] is a widely used and validated inventory assessing masculinity and femininity independently as a two-dimensional construct and therefore determining self-perceived characteristics or traits that have traditionally been expected by men or women. In the present study, a well-validated [54] German version of the short 30-item version by Bem [55] was used (Cronbach’s α = 0.86). For the total of 30 items, participants specify how well the items match their own self-perception on a seven-point Likert scale ranging from 1 (“never or almost never true”) to 7 (“always or almost always true”). Cronbach’s α in the current study for the masculine scale was found to be at 0.89 for men and 0.88 for women and, for the feminine scale, was 0.86 for men and 0.82 for women.

#### 2.2.6. Male Role Norms Scale 

The Male Role Norms Scale (MRNS) [35] is a self-report questionnaire consisting of 26 items with a seven-point Likert scale ranging from 1 (“strongly disagree”) to 7 (“strongly agree”). The MRNS assesses gender ideology and, more specifically, the concept of how a man should be or behave [35]. The items can be summarized to three dimensions of masculinity: status (Cronbach’s α = 0.81), toughness (Cronbach’s α = 0.74), and antifemininity (Cronbach’s α = 0.76) [35,56]. Cronbach’s α in the current study for the individual dimensions was found as follows: for the status dimension, 0.88 in men and 0.81 in women; for the toughness dimension, 0.81 in men and 0.73 in women; and, lastly, for the antifemininity dimension, 0.80 in men and 0.79 in women.

### 2.3. Statistical Analysis

The statistical analysis was carried out in the R software environment for statistical computing and graphics version 4.0.2 (R Core Team 2020) using the additional packages “psych” [57], “rcompanion” [58], and “ggplot2” [59]. The analysis consisted of five steps, where a significance level of α = 0.05 was used throughout all parts. In the first step, means and standard deviations were calculated for the description of the sample (Table 1), and a correlation analysis was conducted to investigate correlations between the used scales for the male and female sample separately. In the second step, Pearson’s two-sided chi-square test was used to determine whether psychotherapy use significantly differed between men and women in the present sample.

In the third step, Student’s two-sided two-samples *t*-test was used to examine the association of gender and psychotherapy use with scores on depression, anxiety and gender role norms. For groups with unequal variances, Welch’s unequal variances *t*-test was used since it is more robust against type I errors in such a case [60]. Four different subgroups, obtained by splitting the sample by gender and psychotherapy use, were directly compared in their mean scores on the aforementioned questionnaires. The aim of this step was to reveal potential differences in the depression, anxiety, gender role identity, and male role norm questionnaire scores among the individual subgroups.

In the fourth step, two-sided binary logistic regression analyses were conducted for male and female participants individually, with self-reported psychotherapy use as the binary outcome variable and scores on the PHQ-9, MDRS-22, GAD-7, BSRI (masculine and feminine subscales), and the MRNS as the predictor variables. The aim of this analysis was to model a potential association of psychotherapy use with the depression, anxiety, gender role identity, and role norm questionnaire scores.

In the fifth step, interaction analyses examining interaction terms between the MRNS and mental health symptoms (PHQ-9, MDRS-22, GAD-7) and gender identity (BSRI-M/-F) were conducted with self-reported psychotherapy use as the binary outcome variable. For this, an interaction term of the two focused variables (e.g., MRNS x PHQ-9) was additionally introduced to the two-sided binary logistic regression model retaining all other variables in the model.

## 3. Results

### 3.1. Descriptive Statistics and Group Differences in Mental Health Symptoms

As presented in Table 1, a total of 716 participants with a mean age of 32.7 (*SD* = 12.2) years reporting to currently suffer from psychological distress were included in the sample. The sample consisted of 37.3% men and 62.7% women. Most of the men reported to feel being masculine (*M* = 2.0, *SD* = 1.4; seven (2.6%) men reported to feel closer to the femininity pole (score ≥ 6) than to the masculinity pole), whereas most of the women felt to be feminine (*M* = 9.0, *SD* = 1.4; nine (2.0%) women reported to feel closer to the masculinity pole (score ≤ 5) than to the femininity pole) (Appendix A). The majority of the participants finished a secondary (54.3%) or a tertiary education (40.1%). Of the participants, 46.5% were single, whereas 30.2% were in a relationship. In the sample, 65.2% of participants identified as heterosexual, whereas 20.1% identified as bisexual and 7.3% as gay/lesbian.

In the present sample, more women (70.3%) than men (29.7%) were undergoing psychotherapy (χ^2^(1) = 7.40, *p* = 0.007). As shown in Figure 1, of all examined men, 66 (24.7%) reported to currently use psychotherapy, whereas, of all women, 156 (34.7%) reported to currently use psychotherapy. Proportionally, men in the current sample as compared to women in the current sample were 29% less likely to use psychotherapy.

Regarding depression symptoms (Figure 2A,B), men showed significantly lower scores on the PHQ-9 (*M* = 13.7, *SD* = 6.0) than did women (*M* = 14.8, *SD* = 5.4) (*t*(714) = −2.53, *p* = 0.011, *d* = −0.20). Depression scores on the MDRS-22 did not differ significantly between men (*M* = 33.7, *SD* = 19.7) and women (*M* = 32.0, *SD* = 17.1) (*t*(498.16) = 1.17, *p* = 0.243). For men in psychotherapy (PHQ-9: *M* = 13.9, *SD* = 6.7; MDRS-22: *M* = 29.8, *SD* = 19.2), depression scores were not significantly different than for men not in psychotherapy (PHQ-9: *M* = 13.6, *SD* = 5.7; MDRS-22: *M* = 35.0, *SD* = 19.7) (PHQ-9: *t*(265) = 0.34, *p* = 0.732; MDRS-22: *t*(265) = −1.88, *p* = 0.062). Women in psychotherapy had significantly higher PHQ-9 scores (*M* = 15.6, *SD* = 5.5) than women not in psychotherapy (*M* = 14.3, *SD* = 5.3) (*t*(447) = 2.50, *p* = 0.013, *d* = 0.25), but their MDRS-22 scores (*M* = 32.3, *SD* = 18.5) did not significantly differ compared to women not in psychotherapy (*M* = 31.9, *SD* = 16.3) (*t*(447) = 0.22, *p* = 0.826).

Concerning the anxiety questionnaire (Figure 2C), men showed significantly lower scores on the GAD-7 (*M* = 10.3, *SD* = 5.0) than did women (*M* = 11.6, *SD* = 4.5) (*t*(507.4) = −3.53, *p* < 0.001, *d* = −0.28). Men in psychotherapy did not have significantly different GAD-7 scores (*M* = 11.2, *SD* = 5.2) than men not in psychotherapy (*M* = 10.0, *SD* = 4.9) (*t*(265) = 1.70, *p* = 0.091). Women in psychotherapy had significantly higher GAD-7 scores (*M* = 12.9, *SD* = 4.0) than women not in psychotherapy (*M* = 10.9, *SD* = 4.6) (*t*(447) = 4.49, *p* < 0.001, *d* = 0.44).

Additional subgroup analyses with only self-identified heterosexual men and women showed the same results in terms of significant group differences between men in psychotherapy and men not in psychotherapy, as well as between women in psychotherapy and women not in psychotherapy (Appendix A).

### 3.2. Correlation Analysis Separated by Gender

As presented in Table 2A, in the male sample, prototypical depression symptoms measured with the PHQ-9, externalizing depression symptoms measured with the MDRS-22, as well as anxiety symptoms measured with the GAD-7 correlated strongly among each other. Masculine role identity assessed by the BSRI masculinity scale correlated negatively with prototypical depression symptoms, externalizing depression symptoms, and anxiety symptoms (Table 2A). The BSRI femininity scale correlated negatively with externalizing depression symptoms. AtTMRN assessed by the MRNS, however, was positively correlated with externalizing depression symptoms but not with prototypical depression symptoms and anxiety symptoms (Table 2A).

As presented in Table 2B, in the female sample, prototypical depression symptoms, externalizing depression symptoms, as well as anxiety symptoms were also strongly correlated among each other. Masculine role identity assessed by the BSRI masculinity scale also correlated negatively with prototypical depression symptoms, externalizing depression symptoms, and anxiety symptoms (Table 2B). As opposed to the male sample, the BSRI femininity scale was not correlated with externalizing depression symptoms, nor was the MRNS correlated with the MDRS-22, but the MRNS and the BSRI femininity scale were negatively correlated among each other (Table 2B).

Using partial correlations that were controlled for age and highest educational level (Appendix A), some minor differences could be observed in the male sample. The correlation between the MDRS-22 and the BSRI masculinity scale did not reach significance anymore (*r* = −0.12, *p* = 0.051), whereas the correlation between the MRNS and the BSRI masculinity scale became significant (*r* = −0.22, *p* = 0.043). All other correlations in the male sample, as well as all correlations in the female sample, did not change their significance or direction of association.

### 3.3. Group Comparisons Regarding Psychotherapy Use and Gender Roles

As presented in Figure 3, men in psychotherapy had significantly lower scores on the BSRI scale assessing masculinity (*M* = 54.2, *SD* = 11.6) compared to men not in psychotherapy (*M* = 59.0, *SD* = 15.3) (*t*(144.95) = −2.71, *p* = 0.008, *d* = −0.33). No difference emerged with regard to the scale assessing femininity in men using psychotherapy (*M* = 67.9, *SD* = 13.2) compared to men not in psychotherapy (*M* = 67.9, *SD* = 13.4) (*t*(265) = 0.03, *p* = 0.977).

Women in psychotherapy did not significantly differ in their scores on the BSRI subscale assessing masculinity (*M* = 54.0, *SD* = 14.2) compared to women not in psychotherapy (*M* = 56.6, *SD* = 13.9) (*t*(447) = −1.82, *p* = 0.070). For women undergoing psychotherapy, there was no significant difference in scores on the BSRI subscale assessing femininity (*M* = 74.6, *SD* = 11.1) as compared to women not in psychotherapy (*M* = 72.8, *SD* = 11.4) (*t*(447) = 1.59, *p* = 0.112).

Considering the MRNS, men had significantly higher scores (*M* = 73.2, *SD* = 26.7) than did women (*M* = 57.7, *SD* = 20.1) (*t*(446.01) = 8.18, *p* < 0.001, *d* = 0.68). Furthermore, men in psychotherapy did not have significantly different MRNS scores (*M* = 70.2, *SD* = 29.1) than men not in psychotherapy (*M* = 74.1, *SD* = 25.9) (*t*(265) = −1.04, *p* = 0.299), but women in psychotherapy had significantly lower scores on the MRNS (*M* = 54.6, *SD* = 16.6) than women not in psychotherapy (*M* = 59.3, *SD* = 21.7) (*t*(392.32) = −2.58, *p* = 0.010, *d* = −0.24).

Subgroup analyses with only self-identified heterosexual men and women showed no differences in regards to the male subsample. For the female subsample, the MRNS scores no longer significantly differed between women in psychotherapy (*M* = 58.1, *SD* = 17.3) and women not in psychotherapy (*M* = 61.7, *SD* = 22.8) (*t*(277) = −1.30, *p* = 0.194) (Appendix A).

### 3.4. Binary Logistic Regression Analysis for the Prediction of Psychotherapy Use

As shown in Table 3 and Figure 4, the binary logistic regression analysis predicting psychotherapy use for male participants was significant (χ^2^ [6] = 20.20, *p* = 0.003, Nagelkerke pseudo *R*^2^ = 0.108). PHQ-9 scores were not significantly associated with psychotherapy use in men. On the other hand, men with higher MDRS-22 scores were significantly less likely to be in the psychotherapy group, and men with higher scores on the GAD-7 were significantly more likely to be in the psychotherapy group (Table 3). Regarding the gender role identity and the male role norm scale, men with higher scores on the BSRI scale measuring masculinity were significantly less likely to be in the psychotherapy group, whereas the BSRI scale measuring femininity as well as the MRNS were not associated with the odds of belonging to the psychotherapy group (Table 3). Repeating the analysis for the male sample with the inclusion of sociodemographic variables (age, relationship status, sexual orientation, and educational level) showed no significant differences. The detailed results for this analysis can be found in the Appendix A.

The binary regression model for female participants was also significant (χ^2^ [6] = 30.05, *p* < 0.001, Nagelkerke pseudo *R*^2^ = 0.089), but showed differing results with regard to the one for male participants, as can be seen in Table 4 and Figure 4. The only significant predictor was the GAD-7, where women with higher scores had significantly increased odds of being in the psychotherapy group (Table 4). Neither the depression questionnaires (PHQ-9 and MDRS-22), nor the BSRI masculinity or femininity measures, nor the male role norms questionnaire were significantly associated with the odds of being in the psychotherapy group (Table 4). Repeating this analysis while controlling for sociodemographic variables (age, relationship status, sexual orientation, and educational level) showed no significant differences. The detailed results for this analysis for the female participants can be found in the Appendix A.

### 3.5. Interaction Analysis Examining Indirect Relations between Adherence to Traditional Male Role Norms and Psychotherapy Use

Adding an interaction term of the MRNS and each mental health questionnaire (PHQ-9, MDRS-22, and GAD-7) separately to the binary logistic regression model for men showed to be an additional significant predictor each time (see Table 3 for the model summaries and Figure 5 for the interaction effects in men). The model with the interaction term MRNS x PHQ-9 (χ^2^(7) = 31.54, *p* < 0.001, Nagelkerke pseudo *R*^2^ = 0.165) showed that the interaction MRNS x PHQ-9 as well as the MDRS-22 and the GAD-7 were significant (Table 3). In this model, the BSRI masculinity scale did not reach significance. The model with the interaction MRNS x MDRS-22 (χ^2^(7) = 27.61, *p* < 0.001, Nagelkerke pseudo *R*^2^ = 0.146) also showed that the interaction term MRNS x MDRS-22 as well as the MDRS-22 by itself were significant, in addition to all the questionnaires that were significant in the model without an interaction term (PHQ-9, GAD-7, BSRI masculinity scale). In the third model, where the interaction term of the MRNS x GAD-7 was introduced (χ^2^(7) = 26.14, *p* < 0.001, Nagelkerke pseudo *R*^2^ = 0.139), the interaction term MRNS x GAD-7 as well as the MDRS-22 and the GAD-7 reached significance. The BSRI masculinity scale was not a significant predictor in this model (Table 3). When the same analysis was conducted while simultaneously controlling for possible covariates (age, relationship status, sexual orientation, and educational level), the results regarding the significance of the interaction terms, as well as the overall model significance, stayed the same in the male sample. Some minor changes were observed in terms of significant predictors in the models with interaction terms, where detailed results can be found in Appendix A.

Including the same interaction terms separately in the logistic regression models for women (see Table 4 for the model summaries and Appendix A for the interaction effects in women) yielded not a single model with a significant interaction term (χ^2^(7) = 36.75, *p* < 0.001, Nagelkerke pseudo *R*^2^ = 0.108). The model with an interaction between the MRNS x PHQ-9 (χ^2^(7) = 32.21, *p* < 0.001, Nagelkerke pseudo *R*^2^ = 0.095) as well as the model with an interaction between the MRNS x MDRS-22 (χ^2^(7) = 33.56, *p* < 0.001, Nagelkerke pseudo *R*^2^ = 0.099) and the model with an interaction between the MRNS x GAD-7 (χ^2^(7) = 31.97, *p* < 0.001, Nagelkerke pseudo *R*^2^ = 0.095) showed no significant interaction. Neither did the model with an interaction between the MRNS x BSRI masculinity (χ^2^(7) = 32.29, *p* < 0.001, Nagelkerke pseudo *R*^2^ = 0.096) nor did the model with an interaction between the MRNS x BSRI femininity (χ^2^(7) = 31.55, *p* < 0.001, Nagelkerke pseudo *R*^2^ = 0.094) yield a significant interaction term. Repeating the same analysis with the same covariates as for the male sample again showed no significant differences in regard to the interaction terms and the overall model significance. Some minor changes could be observed where previously significant predictors no longer reached significance by themselves (detailed results can be found in Appendix A).

## 4. Discussion

### 4.1. Summary of Results

In the present sample of 716 men and women experiencing psychological distress, men were 29% less likely to use psychotherapy as compared to women. Although men exhibited lower levels of prototypical depression symptoms and anxiety symptoms as compared to women, men did not show lower externalizing depression symptoms suggesting similar overall levels of mental health problems. Interestingly, men using psychotherapy did not differ from men not using psychotherapy with regard to depression and anxiety levels, suggesting similar symptom burdens. Women in psychotherapy, however, differed with regard to prototypical depression symptoms and anxiety symptoms from women not in psychotherapy. Women using psychotherapy showed higher prototypical depression and anxiety levels but did not differ with regard to externalizing depression symptoms. Correlational analysis revealed for men a negative correlation between self-identified masculinity with mental health symptoms, whereas self-identified femininity was only negatively correlated with externalizing depression symptoms.

In men, AtTMRN was positively correlated with externalizing depression symptoms but not with prototypical depression symptoms and anxiety symptoms. In women, AtTMRN was neither correlated with externalizing depression symptoms nor with prototypical depression and anxiety symptoms. The binary logistic regression examining potential predictors of psychotherapy use in men revealed a significant model suggesting higher externalizing depression symptoms, lower anxiety symptoms, and higher self-identified masculinity to be associated with a lower likelihood to use psychotherapy. By contrast, in women, only lower anxiety symptoms were associated with reduced psychotherapy use.

Considering the findings of the interaction analyses in men, we found a consistent interaction between AtTMRN and mental health symptoms with regard to psychotherapy use. Men with high AtTMRN indicated psychotherapy use only when concomitantly experiencing high levels of mental health symptoms, whereas in men with low AtTMRN, psychotherapy use was not associated with mental health symptoms.

Additional subgroup analyses for only self-identified heterosexual men and women, as well as supplementary logistic regression analyses controlled for sociodemographic covariates, showed these effects to remain significant even when accounting for the diverse sexual orientations in the present sample.

### 4.2. Integration of Findings

Our finding of reduced psychotherapy use in men by 29% in the current sample is corroborated by a recent nationwide survey from Germany (*N* = 24,016), in which for men, as compared to women, a 28.3% lower utilization of psychiatric and psychotherapeutic services was reported [3]. Previous studies have shown an association between AtTMRN and negative attitudes toward help-seeking in men in general, and for mental health issues in particular [37,38,39,40]. The same studies also suggest AtTMRN to not have the same relevance as an explanatory concept for help-seeking behavior in women as compared to men. The actual use of psychotherapy in individuals experiencing psychological distress and its relation to AtTMRN, gender identity, and externalizing depression symptoms, however, was never examined. The present study revealed higher externalizing depression symptoms, higher self-identified masculinity, and lower anxiety symptoms as factors directly associated with reduced psychotherapy use in men with experienced psychological distress.

It is intuitive that lower anxiety symptoms are associated with a reduced likelihood to engage in psychotherapy. However, the finding of a lower likelihood to engage in psychotherapy with higher externalizing depression symptoms suggests a masked effect of the AtTMRN. Externalizing depression symptoms as well as self-identified masculinity are positively associated with the AtTMRN [43,61]. Admitting to suffer from anxiety does not threaten the masculinity self-concept as much as admitting to suffer from depression [26,41], so that higher levels in the GAD-7 lead to an increased likelihood to engage in psychotherapy in men. However, admitting to suffer from high levels of externalizing depression symptoms reflects an underlying AtTMRN with the two main foci “be in control” and “be unlike women,” which is incompatible with prototypical depression symptoms and psychotherapy use. Traditional concepts of masculinity, based on traditional masculine role norms, are therefore at odds with suffering from depression and psychotherapy use, which are commonly associated with “losing control” and “femininity” [62].

Examining the interaction between AtTMRN and mental health symptoms as previously indicated by Rice and colleagues [43], it emerged that AtTMRN interacts with prototypical and externalizing depression and anxiety symptoms to predict psychotherapy use in men but not in women. Men with high AtTMRN seem to engage only in psychotherapy when experiencing significantly increased levels of mental health symptoms. These findings suggest interaction effects with traditional masculinity as crucial in prediction models of men’s psychotherapy use. With regard to counseling, therapists should be aware of the consequences of high AtTMRN with regard to treatment, namely being associated with more severe symptomatology and probably treatment initiation at later stages of the disease course with a higher likelihood of chronicity of the symptomatology and increased likelihood for suicidal behavior [63].

The finding of a negative association between self-identified masculinity and psychotherapy use in men reflects the manner in which self-identified masculinity is conceptualized and measured. Measured with the BSRI, self-identified masculinity is conceptualized as a set of socially desirable personality traits (e.g., confident, powerful, fearless), which were assumed to be favorable characteristics of men. Self-identified masculinity has been shown to be positively associated with men’s self-confidence [32]. Thus, overall higher masculinity scores in the BSRI might indicate more self-confident and resilient individuals who exhibit a lower need for psychotherapy when experiencing psychological distress. Further, feeling depressed may be in conflict with self-perceived masculine traits, so men who perceive or want to be perceived as having masculine traits are less likely to use psychotherapy in order not to acknowledge feminine traits of being vulnerable or emotional.

The present study stands out clearly from previous studies in this field. A central difference of the present study to previous work is that primarily the attitude towards [38,40] or the intention to start [41] psychotherapy was previously examined and never the actual psychotherapy use.

Another important difference is that, in previous studies, samples were different in the means of not including solely psychologically distressed individuals. Psychological distress is an important criterion for psychological discomfort and may be relevant for the decision to seek help [64,65]. Therefore, in contrast to previous studies that included generally healthy samples not exhibiting increased levels of psychological distress [38,39,40], the present study consisted only of individuals filtered for self-reported psychological distress.

### 4.3. Limitations

There are several limitations that need to be considered while interpreting the results. First, the present study included no clinical diagnostic procedure and assessed psychological distress as well as mental health symptoms by self-report measures. Second, a large portion (*N* = 535, 74.7%) of the sample fell into the age range between 18–40 years, limiting the generalizability to older age groups. Another limitation is the use of gender as a binary construct and in considering only cis-gender-identified people. Additionally, the MRNS was preferred over other similar questionnaires (e.g., Male Role Norm Inventory [66]) because at the time the study was conducted only the MRNS had been validated in a German-speaking sample. 

### 4.4. Future Directions

In future studies, a more extensive and inclusive approach to the self-assessment of gender should be considered, especially since higher prevalence rates of psychological distress have been reported for transgender and gender-diverse individuals because of minority stressors [67,68], and gender role norm measurements seem to not measure the same constructs in trans- and cis-identified persons [69]. Furthermore, although a strength of the present study is the parallel examination of gender identity with the BSRI and male gender ideology with the MRNS, other aspects of masculinity may be relevant with regard to psychotherapy use prediction in men. A more fine-grained analysis of gender identity, adherence or conformity to traditional male or female role norms, and gender role conflict as measured with the Gender Role Conflict Scale [70,71] may provide further valuable insight into the dynamics of the gender gap in psychotherapy use. Furthermore, since this study used a sample of men and women with only self-perceived psychological distress, it would be interesting to conduct similar analyses in subsamples that are comprised of persons with clinically typified psychological distress. Considering that men and women may also differ in regards to the preferences they have for different psychotherapeutic treatment approaches [72,73], future research could also examine these preferences in relation to the traditional gender constructs that were analyzed in the current study. Lastly, there are constructs that might be relevant for actual help-seeking behavior such as stigmatization of mental illness and stigmatization of help-seeking behavior, which further help to better understand help-seeking intentions in men and women with psychological distress [74,75].

## 5. Conclusions

This study provided support to existing research showing lower psychotherapy use in men with mental distress as compared to women with mental distress and linked for the first time actual psychotherapy use in men to lower exhibition of externalizing depression symptoms, lower self-identified masculinity, and higher anxiety symptoms. Furthermore, for the first time, an interaction between AtTMRN and mental health symptoms for the prediction of psychotherapy use in men was reported. Men with high AtTMRN showed an increased likelihood to use psychotherapy when exhibiting elevated mental health symptoms. Such an interaction could not be observed in women. By investigating a mixed sample of men and women, we could further demonstrate crucial gender differences and highlight the need to focus on men’s AtTMRN and its interactions with mental health symptoms for improving psychotherapy uptake in men with psychological distress. Screening for men with high externalizing depression symptomatology and specifically tackling AtTMRN in those men may reduce their reluctance toward psychotherapy and promote psychotherapy uptake earlier in the disease course, preventing chronification of the symptomatology or suicide in men [63]. Finally, psychotherapists might increasingly assess and tackle high AtTMRN in their male patients to resolve psychotherapy-interfering processes originating from high AtTMRN and its incompatibility with psychotherapy.

## Figures and Tables

**Figure 1 behavsci-11-00083-f001:**
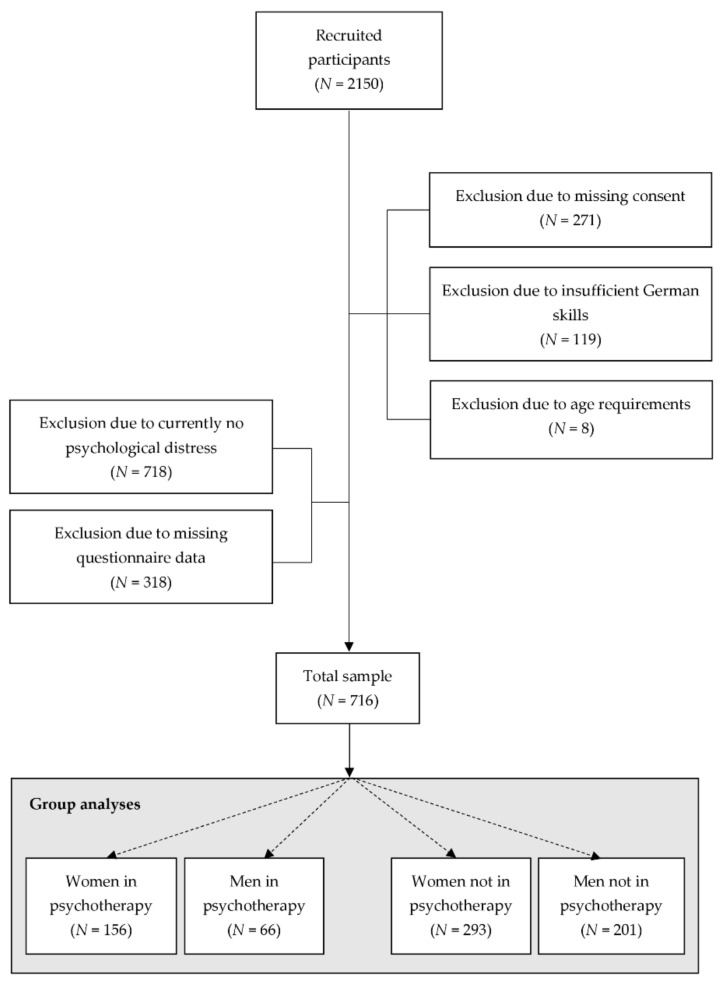
Flow diagram of the inclusion and exclusion process (*N* = number of participants).

**Figure 2 behavsci-11-00083-f002:**
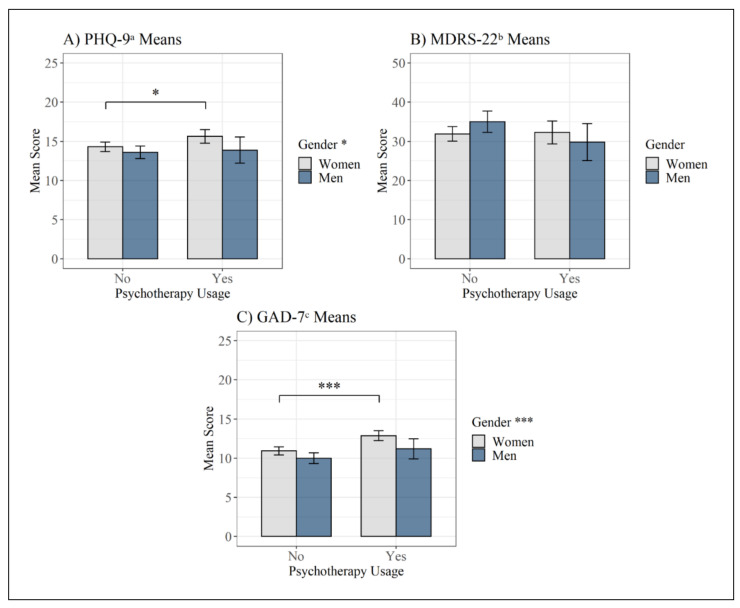
Mean scores on the depression and anxiety questionnaires in the different subgroups with a 95% confidence interval. (**A**) PHQ-9^a^ Means; (**B**) MDRS-22^b^ Means; (**C**) GAD-7^c^ Means. Note: asterisks in the legend indicate significant gender differences regardless of psychotherapy use. ^a^ PHQ-9 = Patient Health Questionnaire-9; ^b^ MDRS-22 = Male Depression Risk Scale-22; ^c^ GAD-7 = Generalized Anxiety Disorder-7; * = *p* < 0.05, *** = *p* < 0.001.

**Figure 3 behavsci-11-00083-f003:**
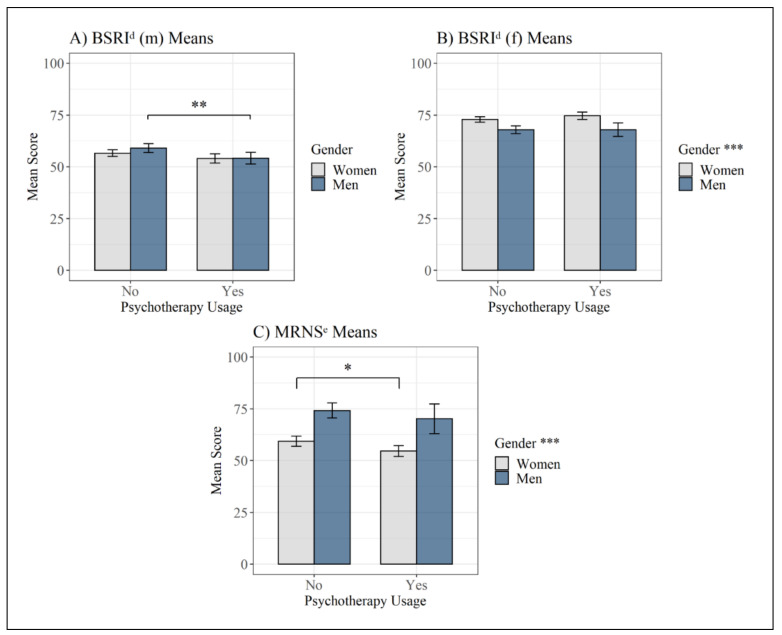
Mean scores on the gender role identity and role norm questionnaires in the different subgroups with a 95% confidence interval. (**A**) BSRI^d^ (m) Means; (**B**) BSRI^d^ (f) Means; (**C**) MRNS^e^ Means. Note: asterisks in the legend indicate a significant difference between the genders, regardless of psychotherapy use. ^d^ BSRI = Bem Sex-Role Inventory; ^e^ MRNS = Male Role Norm Scale; * = *p* < 0.05, ** = *p* < 0.01, *** = *p* < 0.001.

**Figure 4 behavsci-11-00083-f004:**
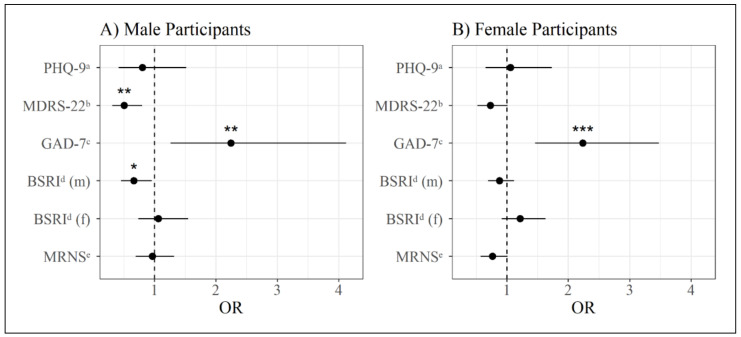
Odds ratios for psychotherapy use based on the questionnaires with a 95% confidence interval. (**A**) Male Participants; (**B**) Female Participants. Note: *m* = masculine subscale, *f* = feminine subscale, *OR* = odds ratio. ^a^ PHQ-9 = Patient Health Questionnaire-9; ^b^ MDRS-22 = Male Depression Risk Scale-22; ^c^ GAD-7 = Generalized Anxiety Disorder-7; ^d^ BSRI = Bem Sex-Role Inventory; ^e^ MRNS = Male Role Norm Scale; * = *p* < 0.05, ** = *p* < 0.01, *** = *p* < 0.001.

**Figure 5 behavsci-11-00083-f005:**
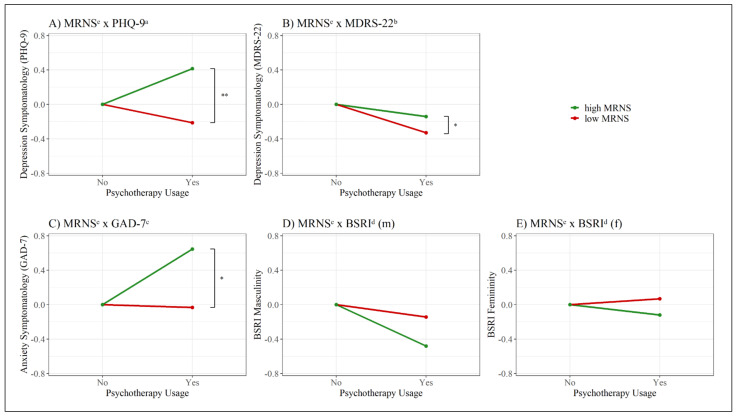
Interaction effects in male participants. (**A**) MRNS^e^ x PHQ-9^a^; (**B**) MRNS^e^ x MDRS-22^b^; (**C**) MRNS^e^ x GAD-7^e^; (**D**) MRNS^e^ x BSRI^d^ (m); (**E**) MRNS^e^ x BSRI^d^ (f). Note: x = interaction, m = masculine subscale, f = feminine subscale. The questionnaire values on the y-axis were z-standardized and linearly transformed to have a joint reference point a y = 0; ^a^ PHQ-9 = Patient Health Questionnaire-9; ^b^ MDRS-22 = Male Depression Risk Scale-22; ^c^ GAD-7 = Generalized Anxiety Disorder-7; ^d^ BSRI = Bem Sex Role Inventory; ^e^ MRNS = Male Role Norm Scale; * = *p* < 0.05, ** = *p* < 0.01.

**Table 1 behavsci-11-00083-t001:** Descriptive statistics for the sample.

Variable	Total (*N* = 716)		In PT (*N* = 222)		Not in PT (*N* = 494)				
	*N (%)*	*M (SD)*	*N (%)*	*M (SD)*	*N (%)*	*M (SD)*	*t (df)*	χ^2^ *(df)*	*p*
Age		32.7 (12.2)		31.9 (11.7)		33.1 (12.4)	−1.25 (449.71)		0.212
Gender								7.40 (1)	0.006 **
Men	267 (37.3)		66 (29.7)		201 (40.7)				
Women	449 (62.7)		156 (70.3)		293 (59.3)				
Education								6.20 (3)	0.102
None completed	7 (1.0)		5 (2.3)		2 (0.4)			3.66 (1)	0.033 *
Secondary education	389 (54.3)		115 (51.8)		274 (55.5)			0.69 (1)	0.407
Tertiary education	287 (40.1)		93 (41.9)		194 (39.3)			0.34 (1)	0.562
Other	33 (4.6)		11 (5.0)		16 (3.2)			0.08 (1)	0.778
Relationship status								0.75 (2)	0.686
Single	334 (46.6)		101 (45.5)		233 (47.2)			0.11 (1)	0.739
In a relationship	327 (45.7)		106 (47.7)		221 (44.7)			0.44 (1)	0.505
Single after a permanent relationship	55 (7.7)		15 (6.8)		40 (8.1)			0.22 (1)	0.638
Sexual orientation								4.33 (4)	0.364
Heterosexual-identified	467 (65.2)		135 (60.8)		332 (67.2)			2.49 (1)	0.115
Gay/Lesbian-identified	52 (7.3)		20 (9.0)		32 (6.5)			1.11 (1)	0.293
Bisexual-identified	144 (20.1)		49 (22.1)		95 (19.2)			0.60 (1)	0.438
Asexual-identified	26 (3.6)		7 (3.2)		19 (3.8)			0.06 (1)	0.808
Other	27 (3.8)		11 (5.0)		16 (3.2)			0.82 (1)	0.367
PHQ-9 ^a^		14.4 (5.6)		15.1 (5.9)		14.0 (5.5)	2.41 (714)		0.016 *
MDRS-22 ^b^		32.7 (18.1)		31.5 (18.7)		33.2 (17.8)	−1.12 (714)		0.265
GAD-7 ^c^		11.1 (4.7)		12.4 (4.7)		10.5 (4.7)	4.86 (714)		<0.001 ***
BSRI ^d^ (m)		56.6 (14.3)		54.1 (13.5)		57.6 (14.5)	−3.04 (714)		0.002 **
BSRI ^d^ (f)		71.4 (12.4)		72.6 (12.1)		70.8 (12.5)	1.81 (714)		0.070
MRNS ^e^		63.5 (24.0)		59.2 (22.2)		65.4 (24.5)	−3.17 (714)		0.002 **

Note: *N* = number of participants, *PT* = psychotherapy, *M* = mean, *SD* = standard deviation, *t* = *t*-statistic, *df* = degrees of freedom, χ^2^ = chi-square test-statistic, *p* = *p*-value, *m* = masculine subscale, *f* = feminine subscale. ^a^ PHQ-9 = Patient Health Questionnaire-9; ^b^ MDRS-22 = Male Depression Risk Scale-22. ^c^ GAD-7 = Generalized Anxiety Disorder-7; ^d^ BSRI = Bem Sex-Role Inventory; ^e^ MRNS = Male Role Norm Scale; * = *p* < 0.05, ** = *p* < 0.01, *** = *p* < 0.001.

**Table 2 behavsci-11-00083-t002:** Correlation matrices for the applied questionnaires.

**(A) Male participants**							
**Variable**	***M***	***SE***	**PHQ-9 ^a^**	**MDRS-22 ^b^**	**GAD-7 ^c^**	**BSRI ^d^ (m)**	**BSRI ^d^ (f)**
PHQ-9 ^a^	13.7	6.0					
MDRS-22 ^b^	33.7	19.7	0.55 ***				
GAD-7 ^c^	10.3	5.0	0.68 ***	0.54 ***			
BSRI ^d^ (m)	57.8	14.7	−0.37 ***	−0.17 *	−0.26 ***		
BSRI ^d^ (f)	67.9	13.3	−0.12	−0.16 *	−0.05	0.33 ***	
MRNS ^e^	73.2	26.7	0.07	0.28 ***	0.11	0.13	−0.21 **
**(B) Female participants**							
PHQ-9 ^a^	14.8	5.4					
MDRS-22 ^b^	32.0	17.1	0.57 ***				
GAD-7 ^c^	11.6	4.5	0.64 ***	0.47 ***			
BSRI ^d^ (m)	55.7	14.0	−0.34 ***	−0.15 *	−0.25 ***		
BSRI ^d^ (f)	73.4	11.3	−0.09	−0.03	0.04	0.26 ***	
MRNS ^e^	57.7	20.1	−0.04	0.06	−0.04	0.04	−0.13 *

Note: *m* = masculine subscale, *f* = feminine subscale, *M* = mean value, *SE* = standard error; ^a^ PHQ-9 = Patient Health Questionnaire-9; ^b^ MDRS-22 = Male Depression Risk Scale-22; ^c^ GAD-7 = Generalized Anxiety Disorder-7; ^d^ BSRI = Bem Sex-Role Inventory; ^e^ MRNS = Male Role Norm Scale; * = *p* < 0.05, ** = *p* < 0.01, *** = *p* < 0.001.

**Table 3 behavsci-11-00083-t003:** Summary of the binary logistic regression models for the male participants.

	No Interaction		MRNS x PHQ-9		MRNS x MDRS-22		MRNS x GAD-7		MRNS x BSRI (m)		MRNS x BSRI (f)	
Predictor	β *(SE)*	*p*	β *(SE)*	*p*	β *(SE)*	*p*	β *(SE)*	*p*	β *(SE)*	*p*	β *(SE)*	*p*
Intercept	0.36 (1.14)	0.750	−1.62 (1.04)	0.119	−1.58 (1.02)	0.122	−1.24 (1.00)	0.212	−1.27 (0.99)	0.198	−1.33 (0.99)	0.181
PHQ-9 ^a^	−0.22 (0.33)	0.499	−0.28 (0.22)	0.198	−0.11 (0.21)	0.610	−0.09 (0.21)	0.685	−0.15 (0.21)	0.479	−0.12 (0.21)	0.556
MDRS-22 ^b^	−0.68 (0.24)	0.005 **	−0.59 (0.20)	0.003 **	−0.80 (0.22)	<0.001 ***	−0.54 (0.19)	0.007 **	−0.56 (0.20)	0.005 **	−0.58 (0.20)	0.004 **
GAD-7 ^c^	0.81 (0.30)	0.007 **	0.96 (0.31)	0.002 **	0.96 (0.32)	0.002 **	0.67 (0.32)	0.034 *	0.79 (0.30)	0.010 **	0.80 (0.30)	0.008 **
BSRI ^d^ (m)	−0.41 (0.19)	0.029 *	−0.35 (0.20)	0.077	−0.39 (0.19)	0.039 *	−0.35 (0.19)	0.063	−0.34 (0.19)	0.082	−0.42 (0.18)	0.024 *
BSRI ^d^ (f)	0.06 (0.19)	0.743	0.05 (0.19)	0.803	0.08 (0.19)	0.681	0.08 (0.19)	0.667	0.05 (0.19)	0.795	0.12 (0.19)	0.544
MRNS ^e^	−0.04 (0.16)	0.818	−0.08 (0.16)	0.611	−0.13 (0.16)	0.413	−0.11 (0.16)	0.506	0.90 (0.55)	0.102	1.01 (0.62)	0.105
Interaction			0.52 (0.17)	0.002 **	0.35 (0.14)	0.011*	0.35 (0.15)	0.016 *	−0.26 (0.15)	0.084	−0.25 (0.15)	0.087
Omnibus statistics												
χ^2^ (*df*)	20.20 (6)		31.54 (7)		27.61 (7)		26.14 (7)		23.42 (7)		23.10 (7)	
*p* (omnibus)	0.003 **		<0.001 ***		<0.001 ***		<0.001 ***		0.001 **		0.002 **	
Pseudo *R*^2^ (Nagelkerke)	0.108		0.165		0.146		0.139		0.125		0.123	

Note: x = interaction, *m* = masculine subscale, *f* = feminine subscale, β = estimated regression coefficient, *SE* = standard error, *p* = p-value. ^a^ PHQ-9 = Patient Health Questionnaire-9; ^b^ MDRS-22 = Male Depression Risk Scale-22; ^c^ GAD-7 = Generalized Anxiety Disorder-7; ^d^ BSRI = Bem Sex-Role Inventory; ^e^ MRNS = Male Role Norm Scale; * = *p* < 0.05, ** = *p* < 0.01, *** = *p* < 0.001.

**Table 4 behavsci-11-00083-t004:** Summary of the binary logistic regression models for the female participants.

	No Interaction		MRNS x PHQ-9		MRNS x MDRS-22		MRNS x GAD-7		MRNS x BSRI (m)		MRNS x BSRI (f)	
Predictor	β *(SE)*	*p*	β *(SE)*	*p*	β *(SE)*	*p*	β *(SE)*	*p*	β *(SE)*	*p*	β *(SE)*	*p*
Intercept	−2.56 (0.81)	0.002**	−2.53 (0.81)	0.002 **	−2.59 (0.81)	0.001 **	−2.69 (0.83)	0.001 **	−2.40 (0.83)	0.004 **	−2.49 (0.83)	0.003 **
PHQ-9 ^a^	0.04 (0.16)	0.817	−0.01 (0.17)	0.948	0.03 (0.16)	0.833	0.04 (0.16)	0.792	0.03 (0.16)	0.868	0.04 (0.16)	0.811
MDRS-22 ^b^	−0.26 (0.14)	0.060	−0.26 (0.14)	0.061	−0.33 (0.15)	0.028*	−0.26 (0.14)	0.063	−0.25 (0.14)	0.067	−0.26 (0.14)	0.064
GAD-7 ^c^	0.81 (0.22)	<0.001 ***	0.82 (0.22)	<0.001 ***	0.81 (0.22)	<0.001 ***	0.84 (0.23)	<0.001 ***	0.80 (0.22)	<0.001 ***	0.79 (0.22)	<0.001 ***
BSRI ^d^ (m)	−0.13 (0.12)	0.286	−0.14 (0.12)	0.259	−0.13 (0.12)	0.276	−0.13 (0.12)	0.293	−0.17 (0.13)	0.184	−0.13 (0.12)	0.280
BSRI ^d^ (f)	0.19 (0.15)	0.188	0.19 (0.15)	0.196	0.20 (0.15)	0.175	0.21 (0.15)	0.163	0.20 (0.15)	0.186	0.18 (0.15)	0.217
MRNS ^e^	−0.25 (0.13)	0.064	−0.24 (0.14)	0.081	−0.25 (0.14)	0.068	−0.28 (0.14)	0.049 *	0.24 (0.53)	0.652	0.04 (0.75)	0.961
Interaction			−0.13 (0.14)	0.374	−0.23 (0.16)	0.153	0.11 (0.15)	0.457	−0.14 (0.14)	0.347	−0.06 (0.15)	0.700
Omnibus statistics												
χ^2^ (df)	31.41 (6)		32.21 (7)		33.56 (7)		31.97 (7)		32.29 (7)		31.55 (7)	
p (omnibus)	<0.001 ***		<0.001 ***		<0.001 ***		<0.001 ***		<0.001 ***		<0.001 ***	
pseudo R^2^ (Nagelkerke)	0.093		0.095		0.099		0.095		0.096		0.094	

Note: x = interaction, *m* = masculine subscale, *f* = feminine subscale, β = estimated regression coefficient, *SE* = standard error, *p* = *p*-value. ^a^ PHQ-9 = Patient Health Questionnaire-9; ^b^ MDRS-22 = Male Depression Risk Scale-22; ^c^ GAD-7 = Generalized Anxiety Disorder-7; ^d^ BSRI = Bem Sex-Role Inventory; ^e^ MRNS = Male Role Norm Scale; * = *p* < 0.05, ** = *p* < 0.01, *** = *p* < 0.001.

## Data Availability

The data used for the present study will be made available by the corresponding author upon request.

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
