# Peer review of "Men’s Psychotherapy Use, Male Role Norms, and Male-Typical Depression Symptoms: Examining 716 Men and Women Experiencing Psychological Distress"

_behavsci, 2021, doi:10.3390/bs11060083_

Round 1
Reviewer 1 Report
Thank you for this beautiful manuscript on an increasingly significant topic. I have a few comments to make.
- In the title, I would replace "usage" with "use."
- Did the authors also measure self-assessed dimensional gender? Many people do not classify themselves as either male or female but feel themselves on a continuum. Was a corresponding variable collected, and can these values be presented for men and women?
- What were the authors' reasons for using the MRNS instead of the MRNI or the CMNI?
- Can the authors repeat the analyses controlling for age, relationship status, sexual orientation, education. It would be interesting to know whether the effects found would hold for these adjustments.
- Regardless of controlling for sexual orientation, a subsample analysis for heterosexual men alone would be important. The literature has already shown that adherence to traditional male role norms differs for heterosexual and homosexual men. Therefore, it would be important to re-examine the findings for heterosexual men alone.
Also, it would be important to control for the above-mentioned covariates in the correlations by using partial correlations for the analysis part.
- An additional point is that the authors did not measure or analyze stigma-related variables, or at least not report it in the manuscript. Did the authors include a variable on stigma in their study? Can the authors report anything on this? If there is such a variable, it would be fascinating to report it in this article. If only part of the sample has data on this aspect, a subsample analysis would be appropriate.
Author Response
A response letter addressing all four reviewers' inquiries is attached.

Reviewer 2 Report
Thank you for the opportunity to review this manuscript. The authors report on a study of psychotherapy usage in a large cross-sectional sample of German-speaking community members who endorse current psychological distress. The aim of the study was to examine whether male-type depression symptoms and adherence to traditional masculinity norms would be related to psychotherapy usage. Analyses were split by gender and included interactions between masculinity norm adherence and symptom severity. The findings are interesting and contribute new information to the field (interactions were significant among men). The manuscript is exceptionally well written, with very clear and precise definition of constructs and good review of extant literature. The methods made sense and analyses were appropriate to the research questions. The authors discussed the implications of the findings well. Overall I strongly endorse this paper. I have only some minor suggestions. First, it could be important to point out, somewhere in the Introduction, that men with more severe externalizing features tend to be at higher risk for suicidal behaviour [Rice SM, et al. J Nerv Ment Dis. 2018;206(3):169-172. doi:10.1097/NMD.0000000000000739]. This just helps underscore how important the present work is, to understand psychotherapy usage among men with externalizing symptoms (and, importantly in this study, adherence to traditional masculinity). Second, when discussing future directions, it might be useful to raise the issue of men’s preferences for different types of psychotherapy and how the constructs in the present study could be examined in relation to men’s preferences for different therapeutic approaches. There are assumptions about this in the literature (e.g., that men want brief instrumental approaches) which do not always hold up under scrutiny. Future research could add to some of the recent work in this area.
Author Response

(The authors gave the same response as above.)

Reviewer 3 Report
This manuscript is a good article, which deserves publication following revision.
The manuscript addresses an exciting topic about gender roles in help-seeking; not seeking help when needed can allow symptoms to increase and other types of manifestations more challenging to treat to occur, even suicide. Therefore, this study is necessary.
I have two suggestions for the authors.
1. I consider it appropriate for the authors to explain in the introduction previous studies on gender roles in help-seeking.
2. I have a question, were the questionnaires used validated in all the samples included? In the negative case, the authors must declare it in the limitations. Different studies have shown that many instruments have different psychometric properties when used in other countries, even though they speak the same language. Many tests have different factorial structures in the same idiom.
Author Response

(The authors gave the same response as above.)

Reviewer 4 Report
A bi-disciplinary team in psychology and psychotherapy and gender medicine of two Universities from Austria presents an analysis on a sample of 716 individuals experiencing psychological distress in a German-speaking population. This is an important topic of gender medicine, with regards to men behavior towards their mental health and adherence to treatments, but also underdiagnosis, that will also benefit from being presented in an open-access to facilitate access to health professionals as well as to society, since it is socially related to Movember movement. Because of this last reason, I would not complain of the long descriptive introduction, as it provides basic conceptual frames of a scenario still poorly taken into account for too many professionals. Still, some suggestion in this respect (move details of literature to discussion) is indicated.
In the next paragraphs, I'd address some issues that still need some clarification or need the authors' attention.
Line 2-4. Title . Since the number of research work addressing 'gender medicine' in such a perspective are not so abundant and the co-first authorship belong to a gender medicine unit I'd suggest to include this concept in the title. At a first glance, I though the paper was focused in males, and only found out it was comparative study in this sentence '
"gender role iden-14 tity, or the experience of prototypical or male-typical externalizing mental health symptoms were 15 associated with psychotherapy usage in men and women."
Impact of gender role identity and roles in psychological distress: worse psychotherapy use and typical depression but not anxiety symptoms in men than women" or something similar.
Line 32: Please, include gender medicine in the keywords.
Lines 72-81. Because of the relevance of the AtTMRN concept, please, put this part as a separated paragraph.
Line 94-128. This is a long body of text that should be divided into two parts. Maybe, the description of works can be moved to the discussion and a short summary sentence can be used here instead and further developed (as it is known, detailing samples, etc) in the discussion.
Lines 144-146 and 146-156. The authors present their hypothesis (144-146), explain the missing gap (146-152) and indicate the goals of the present work (152-156).
This paragraph is the most important, because the aims need to be described and elicit a need to learn more about this topic, filling the gap, is somehow slimed. The challenge, the missing gap " studies focusing on actual psychotherapy usage in psychologically distressed men are 148 lacking. " is clearly defined. So, the next sentence 149-152, and the aims are quite overlapping in the written structure, they are like a paraphrasis and this results in a flattened interest in the description of the goals. I'd suggest omitting 149-152.
Lines 166-167 Participation 166 was limited by age by only including participants between 18 and 100 years old and hav-167 ing good knowledge of the German language.
The concept of limitation looks strange when the whole 'adult' age is covered. I'd say it is the opposite. If one refers to 'adult' mean and women, instead, with no limitation of age (+18-100), would be more.
Why people older than 100 was excluded? Is there a difference between a nonagenarian (99) and a centenarian (101). This can be considered ageist, in the sense that after aprox +65 biological and psychological age do not always correlate with chronological age, and there's no reason to suspect the opposite.
How many of the 8 exclusions in a sample recruited in facebook were aged >100. I guess none?
I'd suggest checking the sample of 718 and see the patterns, as they had no clinically typified 'psychological distress' but they applied, so, one could say they had 'perceived psychological distress and this is also a key question, to be explored, but maybe in a second research work. This is somehow referred to as a point of interest at the end of the discussion when the authors distinguish their work from previous literature by individuals being filtered by "self-reported psychological distress"
Similarly, the analysis of the missing questionnaire data can be interesting to be investigated as it may reflect an avoidance to answer some questions, and probably differ between men and women and with respect to some of their profiles.
Why was gender consider dichotomic and not [men, women, other] considering that the diversity of sexual orientation of the sample will elicit diversity also in the roles?
Since the sample population was identified as diversified regarding sexual orientation, the class 'married' may not apply to all cases of 'cohabitation with a partner??
Lines are lost after 419. Page 16.
Discussion
"716 mentally distressed men and women" please, use '716 men and women with mental distress'. Grammatically both expressions are correct, but in the meaning under an equality perspective, the primary factor is the sex of the person, and afterward, the mental health condition. Some of this 'grammatical' aspects have been strongly claimed in TEA (autism spectrum) but should be applied to all people with any kind of issues regarding mental health.
The same would apply in the discussion for " ….showing lower psychotherapy usage in mentally distressed men as compared to mentally distressed women" --"showing lower psychotherapy usage in men and women with mental distress"
And at " for improving psychotherapy uptake in psychologically distressed men." " for improving psychotherapy uptake in men with psychological distress."
The results in women with/without PT are also interesting. Maybe their findings deserve some space in the focus/aims?
Concerning the sentence " It is questionable whether data from college students or a healthy population is adequate to investigate psychotherapy usage due to psychological distress." It looks lacking impartiality. At least among students, psychological stress is well-known and probably an investigation of psychotherapy use would show lack of time and economic resources playing a role. Please, reconsider the expression 'questionable', mostly considering which is the first limitation stated for the current work.
"Additionally, the use of the MRNS to assess AtTMRN can be criticized as there are measures such as the Male Role Norm Inventory, which show more stable psychometric properties and are more often used in research in English-speaking countries [57]. However, so far there are no other scales translated and validated in German to capture adherence or conformity to traditional male role norms." ------- These two sentences seem to be forced by previous criticisms. The justification is clear, as it is dependent on the validation of that scale into the language used by your sample. So, I'd suggest to start this third limitation in the opposite sense, since the sample was german-speaking, we could not use MRNI, which has better psychometric properties.
" Furthermore, although a strength of the present study is the parallel examination of gender identity with the BSRI and male gender ideology with the MRNS, other aspects of masculinity may be relevant concerning psychotherapy use prediction in men. In future, a more fine-grained analysis of gender identity, adherence or conformity to traditional male or female role norms, gender role conflict as measured with the Gender Role Conflict Scale [58–59] may provide valuable insight into the dynamics of the gender gap in psychotherapy usage."
-------- This part should better be under the subheading of future directions, not limitations
"However, an advantage of the study was that the survey was conducted anonymously further in-creasing participants’ motivation to answer honestly. In addition, no study has ever included only self-reporting psychologically distressed individuals to examine the relation between actual psychotherapy usage, gender-sensi-tive mental health symptoms, gender identity and AtTMRN in parallel." ------ These are positive parts related to the singularity and merit of you work, so I'd suggest finding a place for them in the last sentence of the discussion but not inside mixed with the limitations.
Finally, despite the newly imposed rules about the label to add to the persons attended by the health professionals, I'd like to note that when referring to psychological concern and care, it seems difficult to accept that these persons are considered 'clients' . This is up to the authors' opinion, but I'd like to recall to the hypocritic oath and remind what the client's definition is: A client of a professional person or organization is a person or company that receives a service from them in return for payment.
[business]
...a solicitor and his client.
The company required clients to pay substantial fees in advance.
Synonyms: customer, consumer, buyer, patron
Author Response

(The authors gave the same response as above.)

Round 2
Reviewer 3 Report
The authors have addressed all my comments, and so, the recommendation for acceptance can be given.